# About Time: Do *Transformers* Learn Temporal Verbal Aspect?

**Eleni Metheniti** ① ③
① IRIT (CNRS)
Université Toulouse -
Paul Sabatier (UT3)
31400 Toulouse, France

**Tim Van de Cruys** ① ②
② KU Leuven
Faculty of Arts
Department of Linguistics
Leuven.AI institute
B-3000 Leuven, Belgium

**Nabil Hathout** ③
③ CLLE-CNRS
Université Toulouse -
Jean Jaurès (UT2J)
Maison de la Recherche
31058 Toulouse, France

`firstname.lastname@{univ-tlse2.fr,irit.fr,kuleuven.be}`

## Abstract

Aspect is a linguistic concept that describes how an action, event, or state of a verb phrase is situated in time. In this paper, we explore whether different transformer models are capable of identifying aspectual features. We focus on two specific aspectual features: telicity and duration. Telicity marks whether the verb's action or state has an endpoint or not (telic/atelic), and duration denotes whether a verb expresses an action (dynamic) or a state (stative). These features are integral to the interpretation of natural language, but also hard to annotate and identify with NLP methods. We perform experiments in English and French, and our results show that transformer models adequately capture information on telicity and duration in their vectors, even in their non-finetuned forms, but are somewhat biased with regard to verb tense and word order.

## 1 Introduction

Aspect is a linguistic concept that characterizes how an action, event, or state (expressed by a verb phrase) relates to time, beyond the scope of the verb's tense; via aspect, information such as frequency, duration, and completion is conveyed. Some verbs express events or actions that have or do not have a clearly-defined endpoint because of their meaning (lexical aspect or *aktionsart*), while others can express different temporal properties in different contexts and forms (grammatical aspect). Languages may express aspect in various ways, e.g. by using grammatical verb tense (incomplete actions with continuous/progressive, perfect progressive and imperfect, complete actions with perfect), morphemes (e.g. Finnish, Czech) or with aspect markers (e.g. Mandarin Chinese). However, certain aspectual features cannot simply be deduced from morphosyntax and require some degree of semantic knowledge. In this paper, we focus on two of these aspectual features: telicity and duration. **Telicity** is related to the goal-oriented nature of the verb phrase. The verb's action is said to be *telic* if it has an endpoint; for example, verbs which demonstrate an action such as *kick, eat* ("I kicked the ball.", "I eat an apple.") are *telic*, because the action described has a perceived ending. When the verb denotes a state, e.g. *exist*, or when the completion of the verb's action is either indefinite, impossible or irrelevant, e.g. *agree, stay* ("I agree with you.", "We stayed at the hotel."), then the verb phrase is characterized as *atelic*. **Duration** is another aspectual feature, different from telicity: it distinguishes between verbs that describe a state (*stative*, e.g. *occupy*, *lie*) or an action (*durative*, e.g. *run*, *knock*) regardless of whether they have a perceived endpoint or not. The perception of telicity and duration is the outcome of the entire verbal phrase, and not solely the verb's features (Krifka, 1998). Besides, the context can also place constraints on the aspectual class of a verb (Siegel, 1998). Therefore, making sound judgments on aspectual features such as telicity and duration, especially in a morphologically-poor language like English, is not always an easy task—our datasets in Section 4.1 provide some examples of sentences where these features are hard to assess, even for a human. Aspect has been exploited for tasks where semantic knowledge is necessary, since it provides information on temporal relations (Costa and Branco, 2012), textual entailment (Hosseini et al., 2018; Kober et al., 2019) and event ordering (Chambers et al., 2014).

In recent years, transformer-based models have shown great success in NLP tasks which traditionally require in-depth language analysis and complex strategies on capturing dependencies, semantic information, and world knowledge. However, it remains unclear whether the success of these models is due to a genuine capability to accurately model linguistic meaning, or whether the models are just very good at picking up statistical correlations, but fail to capture fine-grained semantic distinctions (Ettinger, 2020). With this

research question in mind, our goal is to investigate whether transformer-based architectures (both with and without fine-tuning) are able to capture the semantic information related to telicity and duration. To do so, we make use of two datasets annotated for telicity and duration (Friedrich and Gateva, 2017; Alikhani and Stone, 2019), and we conduct a range of experiments using several pretrained transformer architectures in two languages (English and French). We extend our experiments from Metheniti et al. (2021), where we only made use of the Friedrich and Gateva dataset and only in English. We aim to explore the capabilities of transformer architectures in classifying aspect beyond mere quantitative evaluation: we made custom qualitative datasets in order to observe how complex context, verb tense and prepositional phrases affect classification.[1]

We find that classification with fine-tuned models is very successful—both for telicity and duration—and this success can be largely attributed to the knowledge built up during pre-training, as contextual word embeddings by themselves are already quite capable of capturing this information. We noticed that complex cases where the context was conflicting with the verbal aspect were harder for the models to classify, and we provide evidence that misclassification in complex sentences is related to verb tense and word order. Finally, comparing the two languages we investigate, even though the French models show lower accuracy, they were more successful in classifying more difficult cases of telicity and duration, because of the properties of verbal tense in French.

## 2 Acquisition of telicity and duration

Before examining how transformer models handle telicity and duration, it is important to briefly present how humans learn to identify and express these concepts. Complex semantic features are learned by humans with the use of multiple exemplars in the speaker's L1 (mother language), in order to create constructions which encapsulate abstract concepts, such as the perceived duration of an action and the presence or absence of an outcome (Christiansen and Chater, 2001). Frequency (Ellis, 2002) and distributional bias (Andersen, 1993) are crucial for the acquisition of a language's spe-

cific patterns of expressing these concepts, however, their semantics and lexical properties are separate from the grammar of the language and interact with it, to understand and express concepts.

Focusing on lexical aspect, Shirai (1991) and Shirai and Andersen (1995) present the *aspect hypothesis*, claiming that children associate past and perfective marking to telic verbs (applying it to activity, accomplishment and achievement verbs in this order) and avoid such marking with stative verbs. Wulff et al. (2009) confirm this hypothesis experimentally, showing that there is a strong negative correlation between telicity and progressivity (e.g. speakers will mostly avoid using progressive tenses with telic verbs). Todorova et al. (2000) observed, in a self-paced reading experiment, that the combination of aspectually conflicting predicate and temporal modifiers in sentences produced a delay in processing – this suggests that humans have some preferred temporal association with verbs and modifiers, and when there is contradicting context, there is a need for reassessment of the given structure. Proctor et al. (2004) also conducted experiments of self-paced reading, with sentences with verbs whose telicity degree depends—to some extent—on the verb's object (e.g. consumption verbs with a finite/infinite object), and observed that there was no time cost in the processing of these sentences (also pointed out by Todorova et al.), which leads to the conclusion that the processing of a predicate, even with conflicting telicity marking, is simpler than the additional information of a temporal preposition. However, Van Hout (1998) claims that prepositions are mentally learned as markers of telicity earlier in life than the presence of bound/unbound objects (in experiments with Dutch as L1), meaning that some function words are also considered important for the final telicity degree of an utterance.

Regarding duration, in earlier stages of language acquisition, it has been observed that children may erroneously assign stativity to an action without immediate change at the time of utterance (Rocca, 2002), and such mistakes also occur in L2 learners of English (i.e. people who are learning as a foreign language). Wen (1997) also noted that L2 learners of Chinese acquired the perfectivity markers before the duration markers. Such findings further support the *aspect hypothesis*, showing that the perception of time requires a significant amount of processing and contextualizing for humans, and that the lexi-

---

[1] Our code and hand-crafted datasets are made available at `https://github.com/lenakmeth/telicity_classification/`.

cal aspect of a verb (and therefore, the telicity and duration of its presented action/state) is eventually learned and preferred, but can be overwritten (intentionally, in complex cases, at a computational cost, or erroneously, in earlier stages of language acquisition).

## 3 Previous Work

Siegel and McKeown (2000) were the first to propose natural language processing methods for aspectual classification; they used decision trees, genetic programming, and logistic regression to locate linguistic indicators of stativity and completeness, and observed that there was an improvement on the classification of these features, especially with supervised methods, compared to unsupervised classification.

Friedrich and Palmer (2014) use a semi-supervised approach for learning lexical aspect, combining linguistic and distributional features, in order to predict a verb's stativity/duration, and also released two datasets of annotated sentences for stativity. Friedrich and Pinkal (2015) extended this approach by classifying verbal lexical aspect into multiple categories of duration, habitual/episodic/static, and Friedrich et al. (2016) expanded their datasets and categories, achieving 76% accuracy on supervised classification compared to the 80% of their human baseline. In their most recent work, Friedrich and Gateva (2017) have released two datasets in English with gold and silver annotations of telicity and duration (gold is human annotated; silver is obtained from parallel English–Czech corpora where aspectual features were extracted from Czech morphological markers). With these datasets and an L1-regularized multi-class logistic regression model, they report significant improvement on automatic telicity classification.

Loáiciga and Grisot (2016) exploit telicity in order to improve on French–English machine translation; they are using verb classification of telicity (defined as *boundedness*) and notice improvement on the translation of tense. Falk and Martin (2016) also use a machine learning approach, alongside morpho-syntactic and semantic annotations, to predict the aspect of French verbs in different contexts (*verb readings*). Moving away from hard-coded annotations and lexical aspect, Peng (2018) uses two different compositional models to classify aspect, exploring the entire clause and not only the verb, with the use of distributional vectors and without annotated linguistic features, and highlights the importance of the verbal phrase and the verb's dependents in the interpretation of telicity. Kober et al. (2020) propose modeling aspect of English verbs in context, with the use of compositional distributional models, and confirm that a verb's context and closed-class words of tense are strong features for aspect classification.

## 4 Methodology

### 4.1 English Datasets

Telicity and duration-annotated sentences will be used as two separate datasets for our experiments. The two datasets from which we are sourcing sentences are constructed by Friedrich and Gateva (2017) and by Alikhani and Stone (2019).

Friedrich and Gateva's dataset[2] includes gold- and silver-annotations of telicity (telic/atelic) and duration (stative/durative). The gold annotations are based on the MASC dataset (Ide et al., 2008), while the silver annotations were crafted on the basis of the InterCorp parallel corpus of English and Czech (Čermák and Rosen, 2012), extracting the annotations from the Czech morphological markers of telicity and duration and applying them to the English translations. Each annotation corresponds to a specific verb in each sentence and not the entire clause.

The "Captions" dataset[3] by Alikhani and Stone (2019) was created from five image–text corpora, in order to study inferential connections in sentences. It has been annotated for telicity (telic/atelic) and duration (stative/durative/punctual) based on the verb's aspect. Even though the focus of the original work was on the head verb of each sentence, the verbs were not separately annotated, therefore we used dependency parsing with spaCy (Honnibal et al., 2020) in order to extract the verb and its position for our experiments. We noticed some inconsistencies in annotation, which we corrected, and we also excluded the sentences annotated with the *punctual* label, since there were too few sentences to warrant a third category or to combine with the *durative* label.

In Table 1 we present the sizes of the datasets and our final dataset. We split this dataset in training, validation and test sets with a ratio of 80-10-10%.

We also created some smaller datasets for testing

---

[2] https://github.com/annefried/telicity
[3] https://github.com/malihealikhani/Captions

purposes, in order to observe specific phenomena in our models. First, we created forty sentences annotated for telicity, and forty for duration, a sample of which can be found in Table 2. We also crafted "minimal pairs" of sentences with telicity annotations, where each pair includes the same verb but in a context that has a different degree of telicity (see examples in Table 3). We also created variations for some of these sentences, moving prepositional phrases to different positions in the sentence or changing the verb tense without changing the meaning or the degree of telicity, in order to test whether the models are sensitive not only to specific verbs but also word position and tenses (see Table 4).

## 4.2 Verb position

Aspect is generally attributed to the verb; we therefore want to indicate the position of the verb in the sentence. To do so, we make use of a binary mask that indicates the position of the verb form without auxiliaries (or multiple positions, when the verb is split into subwords by the model tokenizer). Technically, we implement the binary mask by making use of so-called `token_type_ids` vectors. These vectors' intended use is to mark tokens of different segments (when performing classification tasks for pairs of sentences)—but since our input consists of a single sentence, we can employ them for specifying the position of the verb. An example is shown in Table 5. Unfortunately, RoBERTa based models (RoBERTa and CamemBERT) do not support the use of `token_type_ids` vectors; we will therefore use these models without an explicit indication of verb position.

## 4.3 Transformer models

Transformers are neural network models which assign weighted attention to the different parts of the input with a sequence of alternating neural feed-forward layers and self-attention layers. These models have proven to be very successful in a variety of NLP tasks, and they have been shown to implicitly capture syntactic and semantic information and dependencies. In this work, we are using pretrained transformer models provided by the `transformers` library (Wolf et al., 2020).

**BERT** (Devlin et al., 2019) is a transformer-based bi-directional encoder, which is trained by randomly masking words in the input sequence and learning to fill the word in the masked position,

| Type | Label | Friedrich | Captions | Current | Total |
|---|---|---|---|---|---|
| **telicity** | telic | 1,831 | 785 | 2,885 | |
| | atelic | 2,661 | 1,256 | 3,288 | **6,173** |
| **duration** | stative | 1,860 | 419 | 2,036 | |
| | durative | 38 | 1,843 | 2,045 | **4,081** |

Table 1: Number of sentences and annotations in each dataset, and our final dataset sizes.

| label | sentence |
|---|---|
| telic | I ate a fish for lunch. |
| telic | John built a house in a year. |
| telic | The cat drank all the milk. |
| atelic | John watched TV. |
| atelic | I always spill milk when I pour it in my mug. |
| atelic | Cork floats on water. |
| stative | Bread consists of flour, water and yeast. |
| stative | This box contains a cake. |
| stative | I have disliked mushrooms for years. |
| durative | She plays tennis every Friday. |
| durative | The snow melts every spring. |
| durative | The boxer is hitting his opponent. |

Table 2: A sample from our qualitative dataset.

| label | sentence |
|---|---|
| telic | I will receive new stock on Friday. |
| atelic | I will receive new stock on Fridays. |
| telic | The boy is eating an apple. |
| atelic | The boy is eating apples. |
| telic | I drank the whole bottle. |
| atelic | I drank juice. |
| telic | The Prime Minister made that declaration yesterday. |
| atelic | The Prime Minister made that declaration for months. |

Table 3: A sample of minimal pairs for telicity.

| label | sentence |
|---|---|
| telic | John built a house in a year. |
| telic | John had built a house in a year. |
| telic | In a year, John built a house. |
| telic | In a year, John had built a house. |
| atelic | We swim in the lake in the afternoons. |
| atelic | We swim in the lake each afternoon. |
| atelic | In the afternoons, we swim in the lake. |
| atelic | Each afternoon, we swim in the lake. |

Table 4: A sample of variations of tense and word order.

| tokens | He | worked | | well | and | earned | | much | | . |
|---|---|---|---|---|---|---|---|---|---|---|
| **vector** | 0 | 1 | | 0 | 0 | 0 | | 0 | | 0 |
| **tokens** | He | work | ###ed | well | and | earn | ###ed | much | . | |
| **vector** | 0 | 1 | 1 | 0 | 0 | 0 | 0 | 0 | 0 | |

Table 5: Sentence tokens and the corresponding `token_type_ids` vectors, depending on tokenization. Each sequence also includes the model's special tokens and padding.

while also learning to predict the next sentence given the first sentence.

**RoBERTa** (Liu et al., 2019) has the same model architecture as BERT, but focuses only on the masked language modeling objective, and expands BERT's use of subwords from unseen words to almost all tokens. The model modifies key hyperparameters in BERT, has been trained with much larger mini-batches and learning rates, and has improved results on the masked language modeling objective and on downstream task performance.

**XLNet** (Yang et al., 2019) is an auto-regressive pretraining model which introduces permutation language modeling, where all tokens are predicted but in random order (unlike BERT, which predicts only the masked tokens). This method allows the model to better learn dependencies and relations between words. XLNet reportedly outperforms BERT on tasks such as question answering, sentiment analysis, and document ranking.

**ALBERT** (Lan et al., 2019) is a transformer architecture, based on BERT but using fewer parameters more efficiently; the vocabulary is decomposed into two small matrices and the size of the hidden layer embeddings (which learn context-dependent representations) is separated from the vocabulary embeddings (which learn context-independent representations). ALBERT has managed to outperform BERT on tasks such as reading comprehension, proving that better exploitation of contextual representations could be more beneficial than larger training and parameter sizes.

### 4.4 Fine-tuning & binary classification

One of our experiments explores the process of fine-tuning a transformer model for binary sequence classification of telicity and duration (separately), and testing the fine-tuned model's accuracy on predicting the telicity or duration annotated label of a sentence. Fine-tuning is the strategy of adapting a pretrained model to a specific task, by adding an extra layer on top of the existing ones and specializing it on the given task. Thus, we can exploit the existing model's knowledge from its contextual word embeddings, and further specialize the model on a specific task without the need for large specialized resources, large computational power and long training times; in many tasks, fine-tuned transformer models have consistently provided state-of-the-art results (Sun et al., 2019).

In order to perform binary classification of telic-ity (telic/atelic) or duration (stative/durative), we first fine-tune the pretrained models on some annotated examples of telicity and duration. The input is entire sentences, with or without the verb position information (presented in Section 4.2), and their label of telicity or duration. We fine-tune the models as Devlin et al. (2019) have recommended, with some modifications; we use a batch size of 32 and a learning rate of $2 \times 10^{-5}$. We apply dropout with probability p = 0.1 and weight decay with $\lambda = 0.01$. We use the PyTorch's ADAM as our optimizer (AdamW) without bias correction. We fine-tune each model for a maximum of 4 epochs, following the recommendation of Devlin et al. (2019) to train for 2-4 epochs when fine-tuning on a specific task. For `base` models each training epoch took ~3 minutes and for `large` models ~7 minutes, on one GPU system of a computing cluster, with CUDA acceleration.

As baselines, we make use of two standard binary classification models trained and tested on the same sets: a simple bag-of-words logistic regression model, implemented with the Python library *scikit-learn* (Pedregosa et al., 2011) with default parameters and data scaling, and a one-layer convolutional neural network model (CNN) implemented with `Pytorch` (Paszke et al., 2019) and trained for 50 epochs, which is commonly used for text classification tasks (Kim, 2014). The CNN model is trained with the fastText 300-dimensional embeddings (Bojanowski et al., 2017), embedding dimension of 300, filter size of $[3, 4, 5]$, 100 filters per dimension, dropout rate of $0.5$, learning rate of $0.01$ and the Adadelta optimizer.

### 4.5 Classification with layer embeddings and logistic regression

Pretrained models already contain linguistic information in their contextualized word embeddings, which we can extract and use with task-specific models for classification. The process of extracting the knowledge of a transformer model's embeddings has been explored since the popularization of contextual word embeddings with ELMo (Peters et al., 2018), since it allows for faster computations with results comparable to fine-tuned transformer models (Tang et al., 2019). We equally conduct an experiment without any finetuning, where we apply a logistic regression to the contextual embeddings of each layer as provided by the pre-trained model. We extract the contextual word embeddings

(for the annotated verb) from each layer of a transformer model, and we train a logistic regression model (using `scikit-learn`) to classify telicity and duration, in order to examine how much information relevant to telicity and duration has been learned by each layer.

## 4.6 Classification in French

We also wanted to examine whether telicity and duration were classifiable in a different language with transformer models. We chose French, as it differs from English in the way verb tenses are formed (conjugation, compound tenses) and used (present continuous is morphologically the same as present simple), but it does not have a dedicated morpheme to expressing telicity such as Finnish and Czech. We are using the two monolingual French transformer models available from the `transformers` library, CamemBERT (Martin et al., 2020) and FlauBERT (Le et al., 2020). CamemBERT is built based on the RoBERTa architecture and trained on monolingual data. FlauBERT is a BERT-based model trained with multiple, heterogeneous corpora, and a more extensive tokenization procedure.

Since there are no available annotations of telicity and duration in French, we translated our English datasets with the DeepL translator[4] and reviewed manually a portion of the datasets (200 sentences) for translation accuracy and annotation correctness. Our average score for the accuracy of the machine-translated sentences was 88% and for the accuracy of the annotated labels was 73.5%. We also extracted the verb-head word of each sentence with the spaCy dependency parser to train with/without verb position, but we are not entirely confident in the results, therefore we are not testing the models' verb embeddings per layer and the unseen verbs of the test set, as we did in English. We use the resulting datasets to fine-tune the FlauBERT and CamemBERT models, and assess their abilities on aspectual classification. In addition, we manually translated our qualitative test sets and made appropriate changes (when verb tense did not convey the desired telicity, for example), and in lieu of the 80 sentences on variations of word order and verb tense, we created more minimal pairs with variations on prepositional phrases.

---

[4]https://www.deepl.com/translator

## 5 Results for English

### 5.1 Quantitative analysis

During the fine-tuning process, we were able to identify via validation which models were most and least successful in predicting binary tags. The results for validation are presented in Table 6 for telicity and Table 7 for duration. In Appendix A.1 we are comparing the probability distributions for the binary labels, for the most successful model (in terms of accuracy).

On classifying **telicity**, the best performing model was `bert-large-cased`. Overall, BERT models outperformed the other architectures, but all models achieved accuracy of $> 0.80$. When trained with the extra information of verb position in the sentence, accuracy improved for all models and sets ($+0.01 - 0.04$). Examining the probability distribution of the two labels, we observed that the BERT models, both `base` and `large`, with the use of the verb position, were the most confident in assigning a label to a sentence (with the probability of each label being $> 0.9$) while the `large` versions of other models were the ones whose probability distribution included more cases with lower label probability. The models were overall more confident with correct predictions, and only very slightly less confident (with a few labels closer to $0.4 - 0.6$, but still the majority above $> 0.9$) for wrong predictions.

Our findings on classifying **duration** were similar to the ones on telicity, with the models performing overall better on this classification task despite the dataset being smaller. The BERT models were the most successful ones, achieving accuracy of up to $0.96$, however all models achieved accuracy of $> 0.93$. The effect of the use of the verb position information is not apparent in this classification task, since we notice an improvement or deterioration of $0.01$ in most models. Examining the probability distribution of the two labels, all models were very confident in classifying sentences, regardless of their accuracy, and high confidence in both right and wrong predictions (erroneously).

In both cases, the fine-tuned transformers models outperformed the baselines we have established.

### 5.2 Qualitative analysis

As mentioned, we also created our own annotated datasets of telicity and duration, in order to study aspectual properties beyond the scope of classification metrics. We took a closer look at the correct

| Model | Verb | Acc. | Prec. | Rec. | F1 |
|---|---|---|---|---|---|
| bert-base-uncased | yes | 0.86 | 0.86 | 0.86 | 0.86 |
| | no | 0.81 | 0.81 | 0.81 | 0.81 |
| bert-base-cased | yes | 0.87 | 0.87 | 0.87 | 0.87 |
| | no | 0.81 | 0.80 | 0.80 | 0.80 |
| bert-large-uncased | yes | 0.86 | 0.86 | 0.86 | 0.86 |
| | no | 0.81 | 0.80 | 0.80 | 0.80 |
| bert-large-cased | yes | **0.88** | **0.87** | **0.87** | **0.87** |
| | no | 0.81 | 0.81 | 0.80 | 0.80 |
| roberta-base | no | 0.84 | 0.84 | 0.84 | 0.84 |
| roberta-large | no | 0.80 | 0.81 | 0.79 | 0.79 |
| xlnet-base-cased | yes | 0.82 | 0.82 | 0.82 | 0.82 |
| | no | 0.81 | 0.81 | 0.81 | 0.80 |
| xlnet-large-cased | yes | 0.82 | 0.82 | 0.82 | 0.82 |
| | no | 0.80 | 0.80 | 0.80 | 0.80 |
| albert-base-v2 | yes | 0.84 | 0.84 | 0.84 | 0.84 |
| | no | 0.81 | 0.80 | 0.80 | 0.80 |
| albert-large-v2 | yes | 0.80 | 0.80 | 0.80 | 0.80 |
| | no | 0.82 | 0.81 | 0.81 | 0.81 |
| CNN (50 epochs) | no | 0.75 | 0.75 | 0.75 | 0.75 |
| Log. Regr. BoW | no | 0.61 | 0.61 | 0.61 | 0.61 |

Table 6: Results of classification accuracy on the telicity test set. 'Verb' refers to training the model with the added information of the verb position.

| Model | Verb | Acc. | Prec. | Rec. | F1 |
|---|---|---|---|---|---|
| bert-base-uncased | yes | **0.96** | **0.96** | **0.96** | **0.96** |
| | no | 0.94 | 0.94 | 0.94 | 0.94 |
| bert-base-cased | yes | **0.96** | **0.96** | **0.96** | **0.96** |
| | no | 0.96 | 0.95 | 0.96 | 0.96 |
| bert-large-uncased | yes | **0.96** | **0.96** | **0.96** | **0.96** |
| | no | 0.95 | 0.95 | 0.94 | 0.94 |
| bert-large-cased | yes | **0.96** | **0.96** | **0.96** | **0.96** |
| | no | 0.95 | 0.95 | 0.95 | 0.95 |
| roberta-base | no | 0.95 | 0.95 | 0.95 | 0.95 |
| roberta-large | no | 0.95 | 0.95 | 0.95 | 0.95 |
| xlnet-base-cased | yes | 0.94 | 0.94 | 0.94 | 0.94 |
| | no | 0.95 | 0.95 | 0.95 | 0.95 |
| xlnet-large-cased | yes | 0.94 | 0.94 | 0.94 | 0.94 |
| | no | 0.95 | 0.95 | 0.95 | 0.95 |
| albert-base-v2 | yes | 0.95 | 0.95 | 0.95 | 0.95 |
| | no | 0.95 | 0.95 | 0.95 | 0.95 |
| albert-large-v2 | yes | **0.96** | **0.96** | **0.96** | **0.96** |
| | no | **0.96** | **0.96** | **0.96** | **0.96** |
| CNN (50 epochs) | no | 0.88 | 0.88 | 0.88 | 0.88 |
| Log. Regr. BoW | no | 0.70 | 0.70 | 0.69 | 0.69 |

Table 7: Results of classification accuracy on the duration test set. 'Verb' refers to training the model with the added information of the verb position.

and incorrect predictions of the models, in order to determine which cases were easier or more difficult for models to classify. For the sake of brevity, we are presenting only a few examples of successes and failures; our goal was to manually examine the strengths and weaknessess of the models in difficult and conflicting cases of classification, hence the smaller qualitative datasets and the presentation of the most interesting examples.

For **telicity**, overall, models were quite successful in classifying the sentences of our qualitative dataset. For example, all models were able to identify that sentences with statements are atelic, such as *Cork floats on water.* and *The Earth revolves around the Sun.*, and sentences with an action were correctly classified almost all the time: *I spilled the milk.* was correctly classified as *telic*, and *I always spill milk when I pour it in my mug.* was also correctly classified as *atelic* (except for the xlnet models).

For the majority of the models, the errors in classification could be located in some specific sentences, where the verb or the verbal phrase would be considered (a)telic, but part of the context defines the temporal aspect of the sentence in the opposite way, either a prepositional phrase (e.g. *I eat a fish for lunch on Fridays.*; *eat* with an object would be considered telic, but the prepositional phrase *on Fridays* shows an action without perceived ending) or a grammatical tense (e.g. *The inspectors are always checking every document very carefully.*; even though the action should have a perceived ending, the continuous tense and the presence of the adverb *always* render this sentence atelic).

Moving to our minimal pairs of telic-atelic sentences, we observe that, in most cases, most models are able to classify correctly a sentence based both on the verb action and the context; *I drank the whole bottle.* and *I drank juice.* were correctly classified as *telic* and *atelic* respectively, despite of the presence of the same verb and tense. However, in our qualitative dataset, we noticed that the sentence *The cat drank all the milk.* was incorrectly classified as *atelic* by all the models. Another interesting mistake we noticed was the classification of the pair *The boy is eating an apple.* and *The boy is eating apples.* as both atelic; in the former sentence, the action is telic for pragmatic reasons (one apple that will be finished), but the tense is continuous.

In order to observe specific tenses, word positions and context more extensively, we can examine the variations of a sentence and see whether the models classified them all with the same label or not. The telic sentence *I ate a fish for lunch at noon.* has confused some of the models, whether the prepositional phrase *at noon* was at the beginning or the end. However, the same sentences regardless of the phrase's position, with past perfect tense *had eaten* is always classified as telic. In some complex cases, such as the sentence *The Prime*

*Minister made that declaration for months.* we notice that most models fail to classify it as atelic in all its variations, except for when the prepositional phrase is at the start and the tense is present perfect continuous (*has been making*). We noticed that even sentences with a more obvious degree of telicity (*John Wilkes Booth killed Lincoln on 1865.* – telic) were sometimes labeled incorrectly, when the prepositional phrase was at the end rather than the start.

Regarding **duration**, the models were less successful at classifying stative sentences than durative; even some sentences with intransitive verbs, such as *Bread consists of flour, water and yeast.* and *This cookbook includes a recipe for bread.* were classified as durative. However, stative sentences with animate subjects such as *I disagree with you.* were correctly classified. Durative sentences, despite of verb tense and context, were always correctly classified, e.g. *She plays tennis every Friday.* and *She's playing tennis right now..*

### 5.3 Layer verb embeddings

By extracting the contextual word embeddings for the verb of each sentence, from each layer, and training a logistic regression model with these embeddings, we were able to examine how much information on telicity and duration is learned by each layer. In Appendix, Figure 3, we present the accuracy for each layer of the *base* models. Models achieved accuracy of up to 79% for telicity classification and up to 90% for duration classification, which is comparable to the performance of the finetuned models. Improvement of accuracy is not proportional as we move to higher layers; we notice that for telicity, some models achieve high accuracy in the middle layers, and again in the final layers, with accuracy sometimes dropping in the last layer.

### 5.4 Unseen verbs

In our training and test datasets, there was a large variety of verb-head words, which allowed us to test the classification success on sentences where the verb has not been observed by the model. For telicity, 267 verb forms which were the head of their phrase were not "seen" by the model in the training set (and 146 of them were not split in subwords), and for duration, 117 verbs (and 80 intact). We tested which of the corresponding sentences were marked incorrectly, and the models' average probability of the assigned label. Overall,

few sentences were labeled incorrectly (see results in Table 10), with labels of either category for both classification tasks. This suggests that the context plays an important role for the models' choices, even when the verb form has not been observed by the model.

## 6 Results for French

### 6.1 Quantitative analysis

The results of the classification for telicity and duration are presented in Tables 8 and 9. Accuracy is overall lower than English, and the CNN classifier baseline performed equally well or sometimes outperformed some models. We questioned whether this was a problem of the machine translation process, but since all sets were created in the same way, we consider this unlikely. However, the fact that the additional verb position information was almost always detrimental is probably a problem caused by parsing, since French makes use of compound tenses more often than English.

| Model | Verb | Acc. | Prec. | Rec. | F1 |
|---|---|---|---|---|---|
| `camembert-base` | no | **0.77** | 0.77 | 0.78 | 0.77 |
| `camembert-large` | no | 0.76 | 0.77 | 0.77 | 0.77 |
| `flaubert-small-cased` | yes | 0.69 | 0.70 | 0.70 | 0.69 |
| | no | 0.73 | 0.73 | 0.73 | 0.72 |
| `flaubert-base-uncased` | yes | 0.74 | 0.75 | 0.74 | 0.72 |
| | no | 0.76 | 0.76 | 0.76 | 0.75 |
| `flaubert-base-cased` | yes | 0.76 | 0.76 | 0.77 | 0.76 |
| | no | 0.77 | 0.78 | 0.78 | 0.78 |
| `flaubert-large` | yes | 0.73 | 0.74 | 0.74 | 0.72 |
| | no | 0.75 | 0.76 | 0.76 | 0.74 |
| `CNN (50 epochs)` | no | 0.71 | 0.69 | 0.65 | 0.65 |
| Log. Regr. BoW | no | 0.61 | 0.59 | 0.59 | 0.59 |

Table 8: Accuracy metrics for telicity classification with French transformer models.

| Model | Verb | Acc. | Prec. | Rec. | F1 |
|---|---|---|---|---|---|
| `camembert-base` | no | 0.82 | 0.82 | 0.82 | 0.82 |
| `camembert-large` | no | **0.87** | 0.87 | 0.87 | 0.87 |
| `flaubert-small-cased` | yes | 0.79 | 0.79 | 0.79 | 0.79 |
| | no | 0.81 | 0.81 | 0.81 | 0.8 |
| `flaubert-base-uncased` | yes | 0.80 | 0.81 | 0.80 | 0.80 |
| | no | 0.84 | 0.84 | 0.84 | 0.84 |
| `flaubert-base-cased` | yes | 0.81 | 0.82 | 0.82 | 0.81 |
| | no | 0.83 | 0.83 | 0.83 | 0.83 |
| `flaubert-large` | yes | 0.81 | 0.81 | 0.81 | 0.80 |
| | no | 0.87 | 0.87 | 0.87 | 0.87 |
| `CNN (50 epochs)` | no | 0.80 | 0.82 | 0.82 | 0.82 |
| Log. Regr. BoW | no | 0.68 | 0.68 | 0.67 | 0.67 |

Table 9: Accuracy metrics for duration classification with French transformer models.

## 6.2 Qualitative analysis

We notice that for French, the fine-tuned models performed better on the qualitative sets than their English counterparts, avoiding common mistakes such as classifying the atelic sentence *Je mange un poisson à midi le vendredi.* ("I eat a fish for lunch of Fridays.") as telic. However, there were (fewer, but some) common mistakes through the models which did not exist for English, e.g. *Je renverse toujours le lait quand je le verse dans ma tasse.* ("I always spill milk when I pour it in my mug." – atelic) and *Jenny a travaillé comme médecine toute sa vie.* ("Jenny worked as a doctor her whole life." – atelic) in which the context affects telicity more than the verb. Comparing minimal pairs, we notice that, unlike in English, the sentence *J'ai bu du jus de fruit.* ("I drank juice." – atelic) was frequently marked as telic by the models, and so did its pair *J'ai bu toute la bouteille.* ("I drank the whole bottle." – telic). And unlike the common mistake of marking both sentences as telic in English, the French models marked the sentences *Le garçon mange [une pomme / des pommes].* ("The boy is eating [an apple / apples]) both as atelic.

For the duration classification, as in English, we observe that stative sentences were the ones which were occasionally or always incorrectly classified by the models; sentences with statements such as *Le pain est composé de farine, d'eau et de levure.* ("Bread consists of flour, water and yeast.") or *J'aime le chocolat.* ("I love chocolate.") were labeled incorrectly.

## 7 Discussion

Transformer models were quite successful in the classification tasks, outperforming our baselines to a large extent, and they proved to be quite successful even without fine-tuning. Contextual embeddings proved to be an efficient way to encode the aspectual information of a verb and its interaction with its context, and this knowledge is probably already learned in the pretraining process.

The superior performance of the duration classification with fine-tuned models did raise a question: from our datasets, most stative questions came from the Friedrich dataset and most durative sentences from the Captions dataset; did the models learn to classify duration or to identify the different corpora? With our qualitative analysis on two languages, we can conclude that the models are indeed able to classify duration and were successful because of the little overlap between stative and durative verbs and contexts. However, the models struggled with sentences for which world knowledge is crucial, which is a known issue (Rogers et al., 2021).

From our experiment with verb tenses and prepositional phrases, we noticed that perfect and continuous tenses are beneficial to classification by the models, and leading a sentence with a prepositional phrase of time sometimes improved predictions. However, conflicting context will almost always confuse the models.

In addition, our findings on the French datasets showed that, even with our lower-performing models, the syntactic and semantic choices that a language makes in expressing aspect did affect the models' capabilities of classifying aspect. The differences in classification errors and successes that we observed, between the qualitative datasets of the two languages, may also indicate that there is a different way in which languages are semantically represented by transformer models, even with different model architectures.

## 8 Conclusion

In this study, we conducted several experiments that test the capability of transformer models to grasp aspectual categories, viz. telicity and duration. We tested this capability using a binary classification setting. Using two annotated datasets for telicity and duration (Friedrich and Gateva, 2017; Alikhani and Stone, 2019), we fine-tuned transformer models of different architectures and in two languages and found that transformer models were very successful on the classification of aspect even when trained on small datasets. Providing the verb position as additional information improved performance in both telicity and duration classification for English. The pretrained transformer models also possess knowledge of aspect even without fine-tuning (when looking at layerwise contextual word embeddings). However, our qualitative analysis also revealed weaknesses; for complex sentences, where the verbal aspect contradicted the temporal information in the context (e.g. telic verb with an atelic prepositional phrase, resulting in an overall atelic sentence), the models classified based on verb rather than context, meaning that they are able to distinguish the most important part of the sequence but not capture more fine-grained information when it is necessary.

## Acknowledgements

This work has been funded by CNRS (80|PRIME-2019 project MoDiCLI). Experiments were carried out with the OSIRIM platform[5] which is administered by IRIT and supported by CNRS, the Region Midi-Pyrénées, the French Government, and ERDF. We would like to thank our reviewers for their insightful comments and suggestions.

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

# A  Additional figures

## A.1  Probability distributions (English)

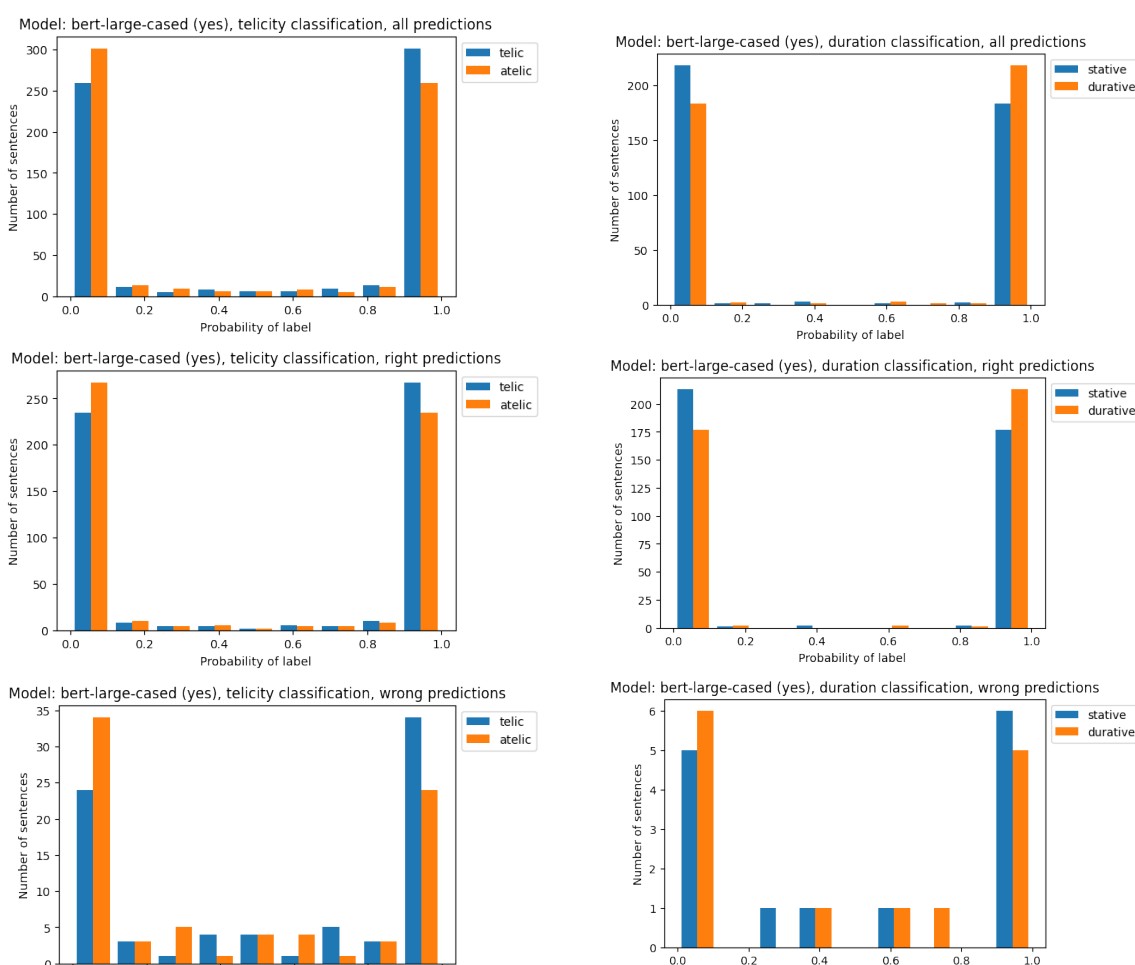

Figure 1: Probability distribution for the telicity labels, for the most successful model (`bert-large-cased` with verb position).

Figure 2: Probability distribution for the duration labels, for the most successful model (`bert-large-cased` with verb position).

## A.2 Correct label predictions on unseen verbs in test set (English)

| Model | Verb | Telicity | | | | | | Duration | | | | | |
|---|---|---|---|---|---|---|---|---|---|---|---|---|---|
| | | Seen verbs | | | Unseen Verbs | | | Seen verbs | | | Unseen Verbs | | |
| | | Correct | Wrong | Acc. | Correct | Wrong | Acc. | Correct | Wrong | Acc. | Correct | Wrong | Acc. |
| bert-base-uncased | yes | 1286 | 240 | 0.84 | 180 | 41 | 0.81 | 681 | 26 | 0.96 | 142 | 6 | 0.96 |
| | no | 1194 | 336 | 0.78 | 170 | 50 | 0.77 | 678 | 29 | 0.96 | 143 | 5 | **0.97** |
| bert-base-cased | yes | 1290 | 218 | **0.86** | 169 | 31 | **0.85** | 665 | 17 | **0.98** | 129 | 5 | 0.96 |
| | no | 1169 | 342 | 0.77 | 162 | 37 | 0.81 | 661 | 21 | 0.97 | 128 | 6 | 0.96 |
| bert-large-uncased | yes | 1292 | 234 | 0.85 | 190 | 31 | 0.86 | 687 | 20 | 0.97 | 142 | 6 | 0.96 |
| | no | 1191 | 339 | 0.78 | 177 | 43 | 0.8 | 688 | 19 | 0.97 | 143 | 5 | **0.97** |
| bert-large-cased | yes | 1308 | 200 | 0.87 | 168 | 32 | 0.84 | 666 | 16 | **0.98** | 128 | 6 | 0.96 |
| | no | 1167 | 344 | 0.77 | 153 | 46 | 0.77 | 667 | 15 | **0.98** | 127 | 7 | 0.95 |
| roberta-base | no | 1243 | 291 | 0.81 | 185 | 41 | 0.82 | 662 | 19 | 0.97 | 126 | 8 | 0.94 |
| roberta-large | no | 1157 | 377 | 0.75 | 176 | 50 | 0.78 | 667 | 14 | **0.98** | 127 | 7 | 0.95 |
| xlnet-base-cased | yes | 1196 | 327 | 0.79 | 174 | 43 | 0.8 | 651 | 30 | 0.96 | 127 | 8 | 0.94 |
| | no | 1175 | 350 | 0.77 | 171 | 45 | 0.79 | 656 | 25 | 0.96 | 129 | 6 | 0.96 |
| xlnet-large-cased | yes | 1190 | 333 | 0.78 | 174 | 43 | 0.8 | 653 | 28 | 0.96 | 127 | 8 | 0.94 |
| | no | 1182 | 343 | 0.78 | 169 | 47 | 0.78 | 652 | 29 | 0.96 | 125 | 10 | 0.93 |
| albert-base-v2 | yes | 1281 | 271 | 0.83 | 186 | 44 | 0.81 | 698 | 16 | **0.98** | 138 | 5 | **0.97** |
| | no | 1194 | 362 | 0.77 | 187 | 42 | 0.82 | 696 | 18 | 0.97 | 137 | 6 | 0.96 |
| albert-large-v2 | yes | 1204 | 348 | 0.78 | 174 | 56 | 0.76 | 690 | 24 | 0.97 | 137 | 6 | 0.96 |
| | no | 1212 | 344 | 0.78 | 184 | 45 | 0.8 | 698 | 16 | **0.98** | 137 | 6 | 0.96 |

Table 10: The results on the test set, for sentences with seen/unseen verbs in the training set, for telicity and duration. The ratio of correct/incorrect labels is similar, with seen and unseen verbs, both for telicity and duration.

## A.3 Classification with pretrained word embeddings and logistic regression (English)

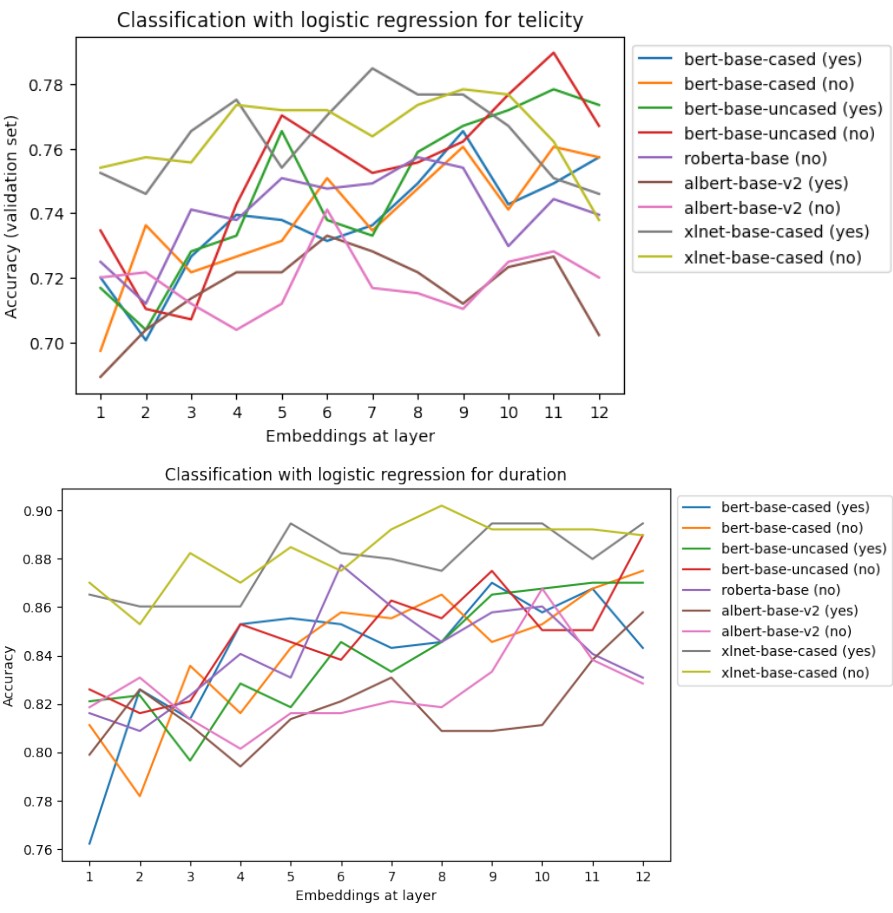

Figure 3: Accuracy of classification of logistic regression, per layer of embeddings, for base models.