# OpenReview forum: "About Time: Do Transformers Learn Temporal Verbal Aspect?"
_aclweb.org/ACL/2022/Workshop/CMCL — CMCL 2022_

### Official Review · Reviewer_776c · 2022-03-21
**A great addition to the workshop**

**Rating:** 9
**Confidence:** 4

**Review:**

This is a very well crafted study, with a good combination of qualitative and quantitative results and a thorough analysis. The conclusions - albeit relatively "obvious in hindsight" - are a good addition to the "BERTology" program and point to some interesting directions for future research. I would have liked to have seen the next step in the analysis, namely, whether verb aspect classification performance is critical to downstream success (i.e. would a Transformer model be able to recover from a misclassification make a correct e.g. event ordering prediction), but this is also a good follow-up topic. Overall, I think this paper is great addition to the workshop.

## Questions to authors
In footnote 4, what does an annotation accuracy of 73.5% refer to? The accuracy of the projected annotations from English? Or the accuracy of the annotator? If it's the former, what was the IAA rate for the whole sample?

Was the finding that BERT had the highest performance for both telicity and duration surprising? Given the (excellently succinct) attribute presentation in §3.3, why weren't the other models able to take advantage of their improvements over BERT?

Do the findings in §4.3 indicate that these models are mostly taking advantage of morphological information? Is that something that could be (or has been) checked with the qualitative datasets?

Could another type of qualitative analysis be constructed with novel (nonse) verbs (Mary daxed carrots all day) and novel contexts (Mary sliced blinkets in foo minutes)?

For the decreased performance of models in French with the verb position information, is parsing accuracy the main culprit (if understand lines 557-8 correctly)? A manual annotation of a small sample of sentences could provide a sanity check for this.

I'm not sure I undersrand the final point in the Discussion section. If the model architectures are different wouldn't it make sense to represent the semantics of language differently?

---

### Official Review · Reviewer_5HRU · 2022-03-23
**A look at aspect in transformers: it's probably there but it's not clear how**

**Rating:** 6
**Confidence:** 4

**Review:**

This paper explores the question of whether sentential embeddings derived from pre-trained language models support the classification of aspect. The paper reports experiments on categorizing two types of aspectual information in English and French using a variety of pre-trained transformer architectures, together with some baseline models. The results point to a positive conclusion, though it is interesting to note that the results in cases where the verb is not explicitly labeled do not represent all that much of an improvement over previous models (Friedrich and Gateva report an F1 of about 76%, which is just a hair lower than most of the no verb cases reported in Figure 10). I appreciated the paper's contribution, though I think it would benefit from a few modifications.

1. I was very interested to read the discussion of the qualitative patterns in the results. However, I found it difficult to extract any general lessons about how the transformers were doing the tasks. The specific deficits that were uncovered in the qualitative analysis could be further probed, by breaking down the quantative results  along a variety of dimensions. For example, it would be revealing to break the results down by aspectual category (e.g., are telic or atelic sentences classified better?), by structure (e.g., are past tense or present tense sentences better classified wrt telicity? what about sentences with definite or indefinite objects?), and by corpus (given that each one was skewed toward one or the other of the categories).  Doing this might allow us to begin to more systematically understand how these complex models are going about the task and why the qualtitative patterns you observed are as they are. I'd like to know for instance how argument definiteness compares as a cue as compared to adverbial type as compared to verb tense as compared to the lexical semantics of the verb.

2. The description of the experiment was not as complete as I would have liked. How was classification done? Specifically, during finetuning, which embedding was used to do classification? Was it the [CLS] token or the verb embedding, and if the latter, what happened in cases where the verb was divided into multiple tokens? Were the embeddings averaged?

3. The discussion of aspect in the paper isn't entirely clear on the distinction between aspect that is lexically determined and that which is the result of structural context (e.g., the properties of the object, adverbial modifiers, or inflectional morphology). The paper doesn't make explicit which of these is being targeted in the classification task discussed in the paper. I would have thought it was the latter, given some of the test cases (in Table 3 we see that "eat" can be either telic or atelic depending on its object). However, some of the comments in the paper lead me to think otherwise: "it has been annotated for telicity and duration based on the verb's aspect" (line 182), "aspect is generally attributed to the verb" (line 215). It would be helpful to clarify this, and to be clear about what diagnostics are being assumed to characterize each of the aspectual categories under discussion. Providing some more detailed characterization of the categories would be particularly helpful in the case of "duration", as it wasn't clear to me what the distinction was -- is it between states and activities (possibly habitual)? Is there a third category of non-durative predicates?

4. I am skeptical that the translation-generated French dataset is sufficiently reliable as to produce interpretable results. Moreover the qualitative assessment didn't reveal any particularly clear generalizations that we can compare with those from English (where the results of the qualitative evaluation were similarly mixed). Moreover, the fact that the results of the pre-trained models was no better than the CNNs makes me worry that they are suffering from a mismatch between the pre-training data and the odd sentences in the aspectual fine-tuning data. Though I am in general quite pleased to see work that compares results across multiple languages, it might not be a bad idea to leave these French results aside until better data sets can be obtained.

Specific comments:

line 167: The procedure for creating the silver annotations in Friedrich and Gateva's dataset strikes me as possibly poblematic. Do we know how reliable it is in producing labels that English speakers agree with? Has this been checked in previous work?

line 226: I don't understand the use of token_type_ids. Is this to specify which embeddings are used for classification? Or does this information get fed into the model so that it facilitate classication? The latter clearly seems like cheating, since we might have expected that the language model would learn about the importance of the verb for aspectual classifcation without any explicit annotation of the verb. And even the former doesn't seem ideal -- shouldn't we be able to classify the entire predicate (or even he entire sentence) as, say, telic or atelic, as compared to the word itself.

Table 1: the differing skew with respect to the duration categories in the datasets is quite striking. As the paper notes, the model could sensibly try to solve this task by determining which corpus it comes from. Do you have a sense that this has happened to any degree? Does it perform worse on durative examples from Friedrich than it does on those from Captions (and vice versa for statives)? Also, why don't the number of labels for telicity equal the number of labels for duration?  Are some sentences only marked for one or the other?

Line 339: What contectual embedding did you use if more than one token is associated with the verb (in the case of subword tokenization)?

Footnote 4: What is the measure of translation accuracy? Entire sentence correctness? Is there a correlation between problematic translations and problematic classification of aspect?

Line 397: Do you have any ideas on why the BERT models yielded more categorical classifcations for aspect (but not duration) as compared to the others?

line 471: Any idea about what made this case difficult?

line 485: Any thoughts on why?

line 494 et seq: Again any thoughts on why?

line 517 (Figure 3): The differences across layers are pretty small. Should we take this to suggest that this classification task can be done using fairly superficial features, such as verb form. Also, it wasn't clear to me whether this probing study was done using the fine-tuned model or not. If not, what do you make of the fact that the results are nearly as good as those obtained from fine-tuning?

line 542: An alternative interpretaion (which doesn't necessarily implicate the use of context) is that the input embedding of the unseen verb is relevantly similar to verbs on which the classifier was trained.

---

### Official Review · Reviewer_s1jw · 2022-03-26
**It performs well on the binary tasks, but has it really learned the distinction?**

**Rating:** 6
**Confidence:** 4

**Review:**

This paper tries to uncover whether transformers are able to distinguish temporal verbal aspect, in both fine-tuned and non-fine-tuned settings on two binary classification tasks related to telicity and duration?

Strength:
The authors look at this problem from multiple perspectives. They created a dataset for French in addition to the available test set for English and create a qualitative dataset to zoom into particular cases.

The experimental set up allows us to see the contribution of several factors individually. For example,  by showing results for unseen vs seen verbs, the contribution of verb-specific knowledge becomes clear. The fine-tuned vs non-fine-tuned set up allows us to see in how far transformers are able to do the task with the general knowledge they contain etc.

Weaknesses:
Even though the experimental design is set up in such a way that we learn quite a bit on how well transformers fare on the task, given several pieces of information, I don’t feel the paper gives an answer to whether transformers actually learn aspectual information. I think this is due to the fact that many tendencies in the results are not explained. For example you mention ‘Examining the probability distribution of the two labels, all models were very confident in classifying sentences, regardless of their accuracy. This to me is a bad sign. The models should be less confident in cases when they are making the incorrect choice if they have actually learned someting. Also, the results on French are worse. The explanation given is not sufficient to me.

I am still afraid that the model just picked up some superficial signals that allow it to do the task wel, without having actually learned the distinction.  Perhaps a simple baseline based on surface cues that the systems could outperform could take away these concerns. Or perhaps the task could be rephrased. Instead of a simple binary decision task, the model could be probed for its understanding of the telicity and durational aspects.

The authors make no specific reference to cognitive theory. It would be nice if they did given the workshop’s focus.

---

### Decision · Program_Chairs · 2022-03-29

Accept